# Virulence Determinants of Colistin-Resistant *K. pneumoniae* High-Risk Clones

**DOI:** 10.3390/biology10050436

**Published:** 2021-05-14

**Authors:** Ozlem Dogan, Cansel Vatansever, Nazli Atac, Ozgur Albayrak, Sercin Karahuseyinoglu, Ozgun Ekin Sahin, Bilge Kaan Kilicoglu, Atalay Demiray, Onder Ergonul, Mehmet Gönen, Fusun Can

**Affiliations:** 1Department of Infectious Diseases and Clinical Microbiology, School of Medicine, Koc University, Istanbul 34450, Turkey; ozldogan@ku.edu.tr (O.D.); cvatansever17@ku.edu.tr (C.V.); natac@ku.edu.tr (N.A.); oalbayrak16@ku.edu.tr (O.A.); oergonul@ku.edu.tr (O.E.); 2Department of Histology and Embryology, School of Medicine, Koc University, Istanbul 34450, Turkey; skarahuseyinoglu@ku.edu.tr; 3School of Medicine, Koc University, Istanbul 34450, Turkey; osahin16@ku.edu.tr (O.E.S.); bkilicoglu15@ku.edu.tr (B.K.K.); ademiray15@ku.edu.tr (A.D.); 4Department of Industrial Engineering, College of Engineering, Koc University, Istanbul 34450, Turkey; mehmetgonen@ku.edu.tr

**Keywords:** colistin resistance, *K. pneumoniae*, iron uptake, net formation, phagocytosis

## Abstract

**Simple Summary:**

The iron uptake systems are associated with virulence in colistin-resistant high-risk clones of *K. pneumoniae*. The isolates belonging to high-risk clones of *K. pneumoniae* are resistant to phagocytosis by neutrophils and induce NET formation.

**Abstract:**

We proposed the hypothesis that high-risk clones of colistin-resistant *K. pneumoniae* (ColR-Kp) possesses a high number of virulence factors and has enhanced survival capacity against the neutrophil activity. We studied virulence genes of ColR-Kp isolates and neutrophil response in 142 patients with invasive ColR-Kp infections. The ST101 and ST395 ColR-Kp infections had higher 30-day mortality (58%, *p* = 0.005 and 75%, *p* = 0.003). The presence of yersiniabactin biosynthesis gene (*ybtS*) and ferric uptake operon associated gene (*kfu*) were significantly higher in ST101 (99%, *p* ≤ 0.001) and ST395 (94%, *p* < 0.012). Being in ICU (OR: 7.9; CI: 1.43–55.98; *p* = 0.024), *kfu* (OR:27.0; CI: 5.67–179.65; *p* < 0.001) and ST101 (OR: 17.2; CI: 2.45–350.40; *p* = 0.01) were found to be predictors of 30-day mortality. Even the neutrophil uptake of *kfu*+-*ybtS*+ ColR-Kp was significantly higher than *kfu*--*ybtS*- ColR-Kp (phagocytosis rate: 78% vs. 65%, *p* < 0.001), and the *kfu*+-*ybtS*+ ColR-Kp survived more than *kfu*--*ybtS*- ColR-Kp (median survival index: 7.90 vs. 4.22; *p* = 0.001). The *kfu*+-*ybtS*+ ColR-Kp stimulated excessive NET formation. Iron uptake systems in high-risk clones of colistin-resistant *K. pneumoniae* enhance the success of survival against the neutrophil phagocytic defense and stimulate excessive NET formation. The drugs targeted to iron uptake systems would be a promising approach for the treatment of colistin-resistant high-risk clones of *K. pneumoniae* infections.

## 1. Introduction

Colistin resistance of *Klebsiella pneumonia* infections is one of the emerging threats in public health because of high fatality rates [1,2,3,4]. The ST101 and ST395 clones of *K. pneumoniae* are known as high-risk clones with high capacity of drug resistance acquisition and are reported to be significant predictors of the mortality [5,6,7,8,9]. The leading virulence factors of *K. pneumoniae* are mostly associated with capsular serotype, muco-viscosity, iron uptake systems, and allantoin metabolism [10,11,12]. Enhanced adhesion and attachment by fimbria and non-fimbrial structures promote pathogenicity of *K. pneumoniae* as well [10,13]. Iron uptake system is essential for survival and dissemination of pathogens during infections. These systems have a significant effect on host inflammatory response [14].

The neutrophils as the important cells of the immune defense, kill pathogens by engulfment or release of extracellular traps (NETs) [15]. The function of NETs is to trap bacteria and promote extracellular killing. This activity may either increase or minimize damage to host cells [15]. A previous study reported that low phagocytic activity of neutrophils contributes to the success of carbapenem-resistant ST258 clone [16]. However, our knowledge on immune escape mechanisms of colistin-resistant *K. pneumoniae* is very limited [11,12]. By this study, we aimed to identify major virulence factors of colistin-resistant *K. pneumoniae* in high-risk clones and show their interaction with the neutrophils. Our results provide an insight to depict the pathogenesis of the colistin-resistant *K. pneumoniae* infection.

## 2. Materials and Methods

### 2.1. Study Population and Data Collection

The patients diagnosed with colistin-resistant *K. pneumoniae* infection between January 2015 and May 2018 from seven healthcare centers in Turkey were included in this study. A study protocol reviewing patient’s demographic data, underlying diseases, type of infection, isolation site, blood biochemical parameters, predisposing factors such as having operation within last 1-month, intensive care unit admission (ICU), type of antimicrobial agents used for empirical and agent-specific therapy, duration of colistin therapy before isolation of colistin-resistant isolates, and carbapenem resistance was used. The patients were followed-up for fatality within 30 days after hospital admission. Exclusion criteria were missing key data, subsequent episodes of the same patient.

### 2.2. Microbiological and Molecular Studies

Colistin resistance was studied by broth microdilution and the breakpoint for resistance was set to >2 mg/L [17]. Carbapenemase genes of OXA-48, NDM-1, and KPC were examined by multiplex-PCR, and amplicons were sequenced [18].

Genotyping of the isolates was carried out by MLST comparing seven housekeeping genes (*phoE*, *gapA*, *rpoB*, *tonB*, *inf*, *mdh*, and *pgi*) according to the protocol published on the Institute of Pasteur website (http://bigsdb.pasteur.fr/klebsiella/klebsiella.html/ (accessed on 7 January 2019)). ST types were determined using Applied Maths Bionumerics V7.6 software.

Virulence genes of type-1 and type-3 adhesins (*fimH-1* and *mrkD*), enterobactin biosynthesis (*entB*), aerobactin receptor (*iutA*), yersiniabactin receptor (*fyuA*), yersiniabactin biosynthesis (*ybtS*), ferric uptake operon associated gene (kfu), regulator of mucoid phenotype A (*rmpA*), capsule type 1 (*magA*), capsule type2 (*K2Wzy*), capsule type 5 (*K5wzx*), outer core lipopolysaccharide biosynthesis (*wabG*), and allantoin metabolism (*allS*) were screened by PCR using primers described previously [19,20,21].

### 2.3. Phagocytosis Assays

For phagocytosis assays, 10 *ybtS*+-*kfu*+, 8 *ybtS*--*kfu*-, 2 *ybtS*--*kfu*+, and 1 *ybtS*+*kfu*- isolates were selected. Phagocytosis assays were performed with slight modifications described by Kobayashi et al. [16]. Colistin-sensitive standard strain *K. pneumoniae* ATCC 700,831 and Gram-positive standard *S. epidermidis* ATCC 35,984 were used as controls. Human neutrophils collected from heathy volunteers were separated from peripheral blood by density gradient centrifugation using Histopaque 1119 (Sigma–Aldrich, Darmstadt, Germany) according to the manufacturer’s instructions. Neutrophil purity was determined by Flow Cytometry (BD Biosciences, Franklin Lakes, NJ, USA)using mouse anti-human CD15-PE (Bechman-Coulter, USA, Indiana) antibody, and above 90% purity was achieved. *K. pneumoniae* isolates were stained with BacLight 488 (Thermo Scientific, Boston, MA, USA) with slight changes to manufacturer’s instructions. For phagocytosis, 2 × 10^7^ neutrophils were incubated with bacterial suspension containing 3 × 10^8^ bacteria for 30 min at 37 °C. Phagocytosis was stopped by adding 1 mL of ice-cold PBS into tubes. A portion of each sample was stained with Mouse-Anti Human CD15-PE (Beckman-Coulter, Indiana, IN, USA) and analyzed with BD Accuri C6 Flow Cytometer. The internalized and/or surface-attached bacteria were determined as CD15+BacLight 488+ cells, whereas free bacteria were determined as only BacLight 488+ Cells. Phagocytic Index (Ph Index) was calculated by the formula ((Initial bacterial count × Ph%)/100). For viability, neutrophils were lysed with dH_2_O for 20 min and cultured on tryptic soy agar by 10-fold dilutions. After overnight incubation, colonies were counted, and the survival index was calculated by ((Colony count per mL/Ph Index) × 100). Experiments were performed in triplicate.

### 2.4. Detection of Neutrophil Extracellular Traps

Two *ybtS*+-*kfu*+ and two *ybtS*--*kfu*- isolates were selected for observation of NET formation. Neutrophils (2 × 10^5^ cells) were incubated for 1 h at 37 °C for attachment to the surface. After incubation, 6 × 10^6^ bacteria were added on neutrophils and incubated 90 min at 37 °C for NET generation (1:30). A portion of each cell was fixed and permeabilized with 4% BSA and 0.2% Triton X-100. After blocking, the cells were stained with Mouse Anti-Human Myeloperoxidase (Santacruz, Heidelberg, Germany) and Rabbit Anti-Human Histone-H3 (Abcam, Cambridge, MA, USA) antibodies for 1 h. Rabbit Anti-mouse Alexa-Fluor 594 (Biolegend, San Diego, CA, USA) and Goat Anti-rabbit Alexa-Fluor 488 (Thermo Scientific, Massachusetts, MA, USA) were used as secondary antibodies. BSA 3% in PBS was used as blocking buffer. Fluoroshield medium with DAPI (Abcam, Cambridge, MA, USA) was used for mounting, and analyses were performed under confocal microscope (Leica DMi8/SP8, Wetzlar, Germany). *K. pneumoniae* ATCC700831 was used as control. NETs were defined as cell-free filaments that were stained with both anti-Histone and anti-MPO antibody. For each slide, 10 microscopic areas were searched by experienced scientists. Degree of NETs formed was categorized into three groups: absence of NETs = 0% neutrophils forming NETs, moderate NET = weak and short NETs by a few neutrophiles, and strong NETs = multiple abundant and long NETs by many neutrophiles per microscopic field. The remaining part was assessed for the viability of the bacteria after NET formation. Cell suspensions were cultured on tryptic soy agar by 10-fold dilutions and colony count per milliliter was recorded.

### 2.5. Statistical Analysis

Statistical analysis was performed using the statistical software package R. In univariate analyses, Wilcoxon rank-sum test for continuous covariates, and Fisher’s exact test for discrete covariates were used. In multivariate analyses, logistic regression was performed using the variables that were detected to be significant in univariate analyses. *p* value below 0.05 was considered as statistically significant. All the results of statistical analysis are available at the Appendix A
https://midaslab.shinyapps.io/klebsiella_pneumoniae_virulence_analysis/ (accessed on 3 November 2020).

Koç University Institutional Review Board approved the study under the number 2015.048.IRB1.008.

## 3. Results

In this study, 142 adult patients with colistin-resistant *K. pneumoniae* infection were included. Overall, 84% (*n* = 119) of the patients stayed in ICU, bacteremia was detected among 43% (*n* = 61) of the patients, and 47% (*n* = 67) of them had ventilator-associated pneumonia (VAP). The median age of the patients was 61 years, and 58% (*n* = 82) of the patients were male. The 30-day mortality was 51% (*n* = 72). Results are shown in Table 1.

The majority of ColR-Kp belonged to ST101 (56%, *n* = 80) and ST395(11%, *n* = 16) clones, and the others (%33, *n* = 46) distributed to various ST clones (minimum spanning tree in supplement). The patients infected with ST101 ColR-Kp had more VAP (*n* = 44.55%, *p* = 0.009) and had higher 30-day mortality rate (*n* = 46, 58% *p* = 0.005) than other clones. The mortality rate among ST395 type *K. pneumoniae*-infected patients was 75% (*n* = 12, *p* = 0.003) (Table 1). The MICs of colistin-resistant isolates were between 4 and 256 mg/L.

Among virulence factors, the presence of ferric uptake operon associated gene (*kfu*) and yersiniabactin (*ybtS*) components of iron uptake systems were found to be significantly higher in ST101 and ST395 ColR-Kp compared to the other clones. The ybtS and kfu positivity were 99% in ST101 (79/80, *p* ≤ 0.001) and 94% in ST395 clones (*n* = 15/16, *p* < 0.012). The mucoid type associated gene (*rmpA*) and *fimH* type adhesin were also significantly higher in ST101 with the percentage of 89% (*n* = 71/80, *p* = 0.005) and 99% (*n* = 79/80, *p* = 0.024), respectively.

The carriage of OXA-48 carbapenemase was significantly higher (*n* = 76/80, 95%) in ST101 than the other clones (76%), (*p* = 0.003). However, it was found to be significantly lower (*n* = 5/16, 31%) in ST395 clone than ST10, (*p* = 0.002). On the contrary, NDM-1 production was significantly higher in ST395 (*n* = 14/16, 88%, *p* < 0.001) and lower in ST101 (*n* = 3/80, 4%, *p* < 0.001) than the other clones (30%) (Table 2). Three ST395 strains were found to be positive for OXA-48 and NDM-1. None of the colistin-resistant *K. pneumoniae* isolates carried KPC-type carbapenemase.

In univariate analysis, being in ICU (OR: 4.3; CI: 1.42–16.04; *p* = 0.005), presence of *ybtS* (OR: 3.0; CI: 1.01–10.02; *p* = 0.034) and *kfu* (OR: 3.9; CI: 1.27–14.63; *p* = 0.009) were found to be associated with 30-day mortality. In multivariate analysis, being in ICU (OR: 7.9; CI: 1.43–55.98; *p* = 0.024), *kfu* (OR: 27.0; CI: 5.67–179.65; *p* < 0.001) and ST101 (OR: 17.2; CI: 2.45–350.40; *p* = 0.01) were found to be the predictors of 30-day fatality (supplement).

The phagocytosis experiments showed that *ybtS*- and *kfu*-positive ColR-Kp were internalized at higher rates (median = 78%), while negative isolates exhibited low phagocytosis rates (median = 65%) after 30 min of interaction with neutrophils (*p* < 0.001, Figure 1). The phagocytosis rates of *S. epidermidis* and *K. pneumoniae* ATCC controls were 89% and 73%, respectively. Survival of *kfu*+-*ybtS*+ positive ColR-Kp was significantly higher than negative isolates with median survival index of 7.90 (range: 3.29–13.13) vs. 4.22 (range: 0.36–5.64), respectively (*p* = 0.001). The survival index of *S. epidermidis* was detected as 0.64, and it was 1.89 for *K. pneumoniae* (Figure 2). The survival index of two *ybtS*--*kfu*+ isolates were 12.04 and 12.13, and it was 5.13 in one *ybtS*+-*kfu*- isolate. Among all isolates studied by phagocytosis assay, four was in ST101 clone. The median phagocytosis rate was found to be 80%, and the survival index was 8.51.

After NET formation, the mean colony count of two *kfu*+-*ybtS*+ isolates was 5.50 × 10^6^, and it was 4.05 × 10^6^ for *kfu*--*ybtS*- strains, while the colony count of ATCC *K. pneumoniae* was 4.3 × 10^6^. Confocal microscopy images showed that the *kfu*+-*ybtS*+ isolate stimulated strong NET formation with excessive release of chromatin granular content to the extracellular area in each microscopic field. However, the *kfu*--*ybtS*-negative ColR-Kp caused moderate NETs seen only in a few areas. NETs were absent on the slide of control ATCC strain (Figure 2).

## 4. Discussion

Infections with high-risk clones of *K. pneumoniae* are usually fatal since therapeutic options are limited due to extensive drug resistance and successful immune escape mechanisms of these pathogens. Alternative approaches are urgently needed in order to prevent and treat infections. Likewise, one of the most promising strategies is inhibition of virulence factors of the bacterium. Here, we revealed the major virulence factors of *K. pneumoniae* high-risk clones ST101 and ST395 and found an enhanced neutrophil activity against the isolates carrying iron uptake system-related genes.

In this study, a significantly higher presence of iron uptake associated gene (*kfu*) and yersiniabactin (*ybtS*) positivity were observed in ST101 (99%) and ST395 (94%) isolates (*p* < 0.001 and *p* = 0.012, respectively). Holden et al. demonstrated that during pneumonia, siderophores stabilizes hypoxia inducible factor (HIF-1alpha), which controls vascular permeability and increases bacterial dissemination to the spleen [14]. Yersiniabactin and iron uptake genes were found to be associated with high mortality and dissemination of infections [22,23].

The *kfu* gene is also associated with higher virulence in bacterial infection murine model [24]. In our study, multivariate analysis showed that *kfu* predicts 30-day mortality (OR: 27; CI: 5.67–179.56; *p* < 0.001) and is a predictor of belonging ST101 clone (OD:20.3; CI: 2.17–484.56; *p* = 0.018). Lawlor et al. reported that the acquisition of yersiniabactin is an important step in the evolution of virulent *K. pneumoniae* [2]. Therefore, we postulated that iron uptake systems in high ST101 and ST395 risk clones of *K. pneumoniae* could be responsible of high mortality rates of infections caused by these clones.

Another important disease strategy of hypervirulent clones is immune evasion from the innate response [25,26]. Interaction of siderophores with host cells promotes pathogenicity of *K. pneumoniae* by induction of proinflammatory cytokines [23]. Proinflammatory cytokines have a protective effect against *K. pneumoniae* by recruitment of neutrophils to the infection site. However, studies pointed out the evasion strategy of virulent *K. pneumoniae* through yersiniabactin secretion [16,23,24,27,28]. One important effect of yersiniabactin is evasion from innate immune protein Lipocalin 2, which is produced by neutrophils or mucosal surfaces [24]. The other effect of yersiniabactin is the enhancement of bacterial survival in phagocytic cells by reduction in the oxidative stress response [27]. We proposed that although neutrophils could intake the *kfu*+-*ybtS*+ producing ColR-Kp, the high survival rates of bacteria were probably due to bacterial resistance to intracellular killing processes of neutrophils. Capsular polysaccharides of ST258 clone of *K. pneumoniae* were reported to have an inhibition on phagocytosis activity of neutrophils [16]. In this study, we did not find a difference in capsule types of high-risk clones and others.

Another significant finding of our study was NET release from neutrophils after encountering *kfu*+*ybtS*+ ColR-Kp (Figure 2). We observed the extensive spread of myeloperoxidase and histone in the extracellular space of neutrophils. The role of NETs in the infection pathogenesis is still under debate. While some pathogens are killed by NETs, others may survive or even benefit from NETs [29]. Phagocytosis is a critical event of decisions to form NETs, and if the bacterium is killed by phagocytosis, only a few azurophilic granules may leak into extracellular space with no NET formation [30]. Similarly, the *kfu*--*ybtS*- isolates of our study induced very rare NETs with a low amount of myeloperoxidase and histones in extracellular space (Figure 2). As a control of our experiment, we studied *K. pneumoniae* ATCC strain and we did not observe NET formation. Branzk et al. reported that virulent bacteria may circumvent phagocytosis by the formation of large aggregates and trigger NET formation [30]. The successful survival of *kfu*+-*ybtS*+ isolates (median survival index 7.9) from phagocytosis with induction of extensive formation of NETs suggested us that the protective function of iron uptake systems from being killed by neutrophils might be one of the reasons for mortality of the patients through increased inflammation.

Overall, the 30-day mortality was 51% and being in ICU was found to be significantly associated with the 30-day mortality. Bacteremia was detected among 43% of the patients and 47% of them had ventilator-associated pneumonia (VAP). *K. pneumoniae* ST101 is known as hypervirulent clone mostly responsible for pneumonia and bacteremia in intensive care units. [7,8]. This clone was also presented as dual-risk clone because of carriage of antibiotic resistance genes and several known virulence factors. In a recent study, the ST101 genetic repertoire was defined as a “perfect storm” allowing for newly emerging resistance and virulence genes [31]. In this study, ColR-Kp ST101 isolates were found to be associated with VAP infections (*p* = 0.009). The ST395 clone is known as a potentially high-risk clone [4], and recent studies pointed out the emergence of carbapenem-resistant ST395 in France and Italy [32,33]. KPC-2 producing ST101 *K. pneumoniae* was shown to have the highest number of virulence genes associated with capsule type, attachment and iron uptake compared to other epidemic clones of *K. pneumoniae* [11]. Similarly, we observed higher presence of virulence genes for iron uptake system, attachment and mucoid phenotype among isolates belonged to ST101 and ST395 than other clones (heatmap, supplement).

The weak part of the study is, we did not confirm the correlation of iron uptake gene positivity and neutrophil response by further genetic studies. Our data represent the results of observational analyses. Our novel findings in depiction of pathogenesis of strains that are referred as high-risk clones should be supported by the animal studies.

## 5. Conclusions

In conclusion, iron uptake systems have a significant contribution to the pathogenesis hypervirulent *K. pneumoniae* ST101 and ST395 infections. These systems contribute to successful survival of *K. pneumoniae* against neutrophil phagocytic defense and stimulate NET formation. The drugs targeted to ferric uptake systems would be a promising approach for treatment of hypervirulent *K. pneumoniae* infections.

## Figures and Tables

**Figure 1 biology-10-00436-f001:**
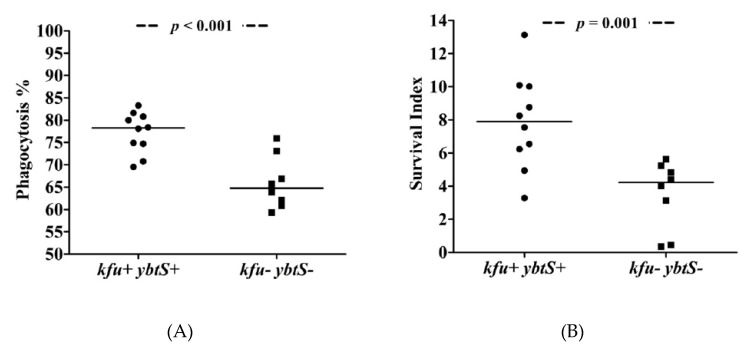
The phagocytosis of ColR-Kp by neutrophils. Phagocytosis rate of *kfu*+-*ybtS*+ and *kfu*--*ybtS*- isolates (**A**) and survival of *kfu*+-*ybtS*+ and *kfu*--*ybtS*- isolates after being phagocytosed by neutrophils (**B**).

**Figure 2 biology-10-00436-f002:**
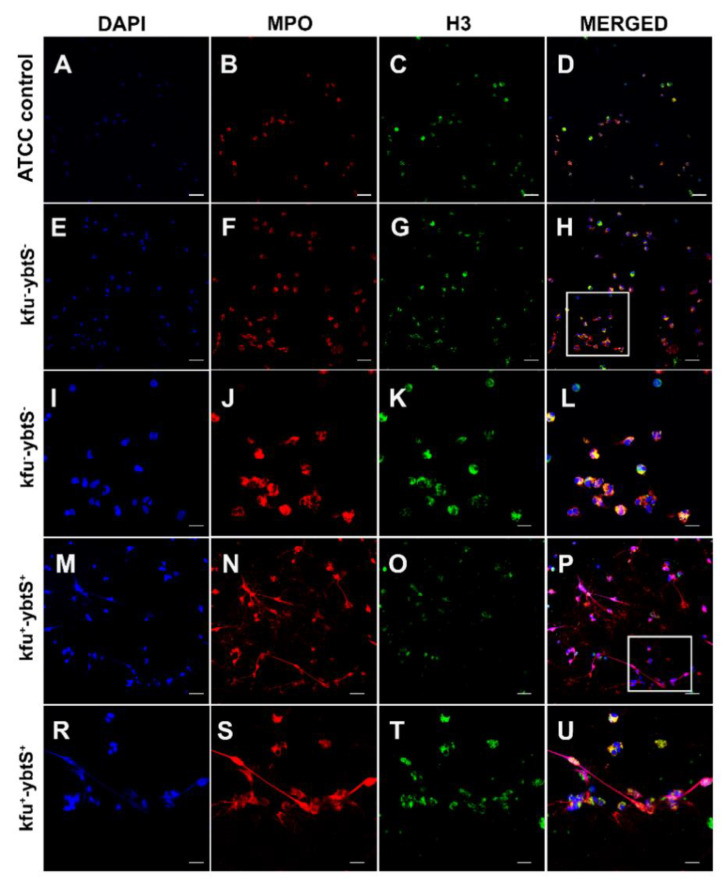
Confocal microscopic images of NETs. The samples were stained consecutively with myeloperoxidase (MPO, red) and histone 3 (H3, green). The nuclei were counterstained with DAPI (blue). Neutrophils were seen intact with *K. pneumonia* ATCC 700,831 control (**A**–**D**). The *kfu*- -*ybtS*- isolates depicted rare and weak NET formation (**E**–**L**). The rectangular area in image H was magnified in images **I**–**L**. The *kfu*+ -*ybtS*+ isolates showed abundant NET formation with excessive histone and MPO release in extracellular matrix (**M**–**U**). The rectangular area in image *p* was magnified in images **R**–**U**. Bars: **A**–**H**, M-*p* = 25 μm; **I**–**L**, **R**–**U** = 10 μm.

**Table 1 biology-10-00436-t001:** Clinical characteristics of the patients infected with colistin-resistant *K. pneumoniae*.

Patient	Total (*n* = 142) *n* (%)	ST101 *n* = 80 *n* (%)	ST395 *n* = 16 *n* (%)	Others * (*n* = 46) *n* (%)
Age Median (range)	61 (0–91)	63 (0–86)	62 (30–84)	53 (0–91)
Female gender	60 (42)	34 (43)	7 (44)	19 (41)
Bacteremia	61 (43)	33 (41) *p* = 0.852	10 (63) *p* = 0.147	18 (39)
VAP	67 (47)	44 (55) *p* = 0.009	9 (56) *p* = 0.079	14 (30)
30-day mortality	72 (51)	46 (58) *p* = 0.005	12 (75) *p* = 0.003	14 (30)
Being in ICU	119 (84)	68 (86) *p* = 0.323	15 (94) *p* = 0.261	36 (78)

* Non-ST101 and Non-ST395 ColR-Kp. *p* values indicate ST101 vs. others and ST395 vs. others. *p* < 0.05 was accepted as statistically significant.

**Table 2 biology-10-00436-t002:** The virulence factors and carbapenamase types in the colistin-resistant *K. pneumoniae* ST101 and ST395 clones.

	Mucoid Type and Capsule *n* (%)				Iron Metabolism *n* (%)					Adhesins *n* (%)		LPS *n* (%)	Allantoin *n* (%)	Carbapenemase *n* (%)	
	*rmpA*	*magA*	*K2wzy*	*K5wzx*	*fyuA*	*kfu*	*IutA*	*ybtS*	*entB*	*mrkD*	*fimH*	*wabG*	*allS*	OXA-48	NDM-1
ST101 *n* = 80	71 (89)	6 (8)	31 (39)	0	79 (99)	79 (99)	4 (5)	79 (99)	80 (100)	79 (99)	79 (99)	80 (100)	0	76 (95)	3 (4)
*p*	0.005	1	0.707	-	0.553	<0.001	0.285	<0.001		0.059	0.024	0.365	-	*p* = 0.00	*p* < 0.001
ST395 *n* = 16	14 (88)	0	9 (56)	0	15 (94)	15 (94)	1 (6)	15 (94)	15 (100)	15 (94)	16 (100)	16 (100)	0	5 (31)	14 (88)
*p*	0.194	0.565	0.401	-	1	0.012	1	0.012		1	0.315	1	-	*p* = 0.002	*p* < 0.001
Others *n* = 46 *	32 (68)	23 (49)	20 (43)	0	45 (96)	28 (60)	5 (11)	28 (60)	47 (100)	43 (92)	42 (89)	46 (98)	0	36 (77)	14 (30)

* Non ST101 and NonST395 ColR-Kp. *p* < 0.05 was accepted as statistically significant.

## Data Availability

The data are available at https://www.biorxiv.org/content/10.1101/677492v1.full.pdf (accessed on 5 May 2021).

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
