# Peer review of "Virulence Determinants of Colistin-Resistant K. pneumoniae High-Risk Clones"

_biology, 2021, doi:10.3390/biology10050436_

Round 1
Reviewer 1 Report
Major comments
In fact the authors say that the "data represent the results of observational analyses", despite being an observational study, could be well discussed. The discussion is very incomplete, the justification is presented with phrases taken from the articles and out of context, they do not validate the results.
1-The authors introduce a statistical process to justify, (validate) the results, however, the p-value could be equal 0.852 or 0.003 or p<0.001. What is value-added and how does it help the data validation? and it was not discussed.
2-The presentation of the results in % must be preceded by the number, or vice-versa example in line 132
142 (n=28, 20%)
142 (20%, n=28)
Minor comments
1- Lines 132-136. these results expressed in table 1 must be indicated
2-Line 137- "The majority of ColR-Kp belonged to ST101 (56%) and ST395(11%) clones" All 142 strains were ColR-Kp? so 20% were adult patients and 80% were children?
3-Lines 149-150 "significantly higher (95%, p=0.003)" and "very low (31%, p=0.002)" could you explain the meaning of p-values?
4- Lines 150-151-Are there three ST395 strains with two carbapenemases?
5- Line 191 "was magnified in images I-L" must be R-U
6- lines200-201 "One of the most promising strategies is inhibition of virulence factors of bacterium. one of the examples mentioned in major comments
Lines 209-210 "Holden et al. demonstrated that during pneumonia siderophores stabilizes HIF-1 and increases bacterial dissemination to the spleen [14]. what is the meaning of HIF-1? and explain this sentence.
lines 211-212 "Lawlor et al. reported that the acquisition of yersiniabactin is an important step in the evolution of virulent K.pneumoniae [2,24]" and Bachman, M.A. [24]!!!!
Lines 226-227 "ColR-Kp could be explained by their resistance inside neutrophils after being uptake". explain this sentence.
Line240- "Branzk et al. reported that NETs are formed in response to large pathogens" is similar to title of article!!! explain this sentence.
Author Response
Comments and Suggestions for Authors
Major comments
In fact the authors say that the "data represent the results of observational analyses", despite being an observational study, could be well discussed. The discussion is very incomplete, the justification is presented with phrases taken from the articles and out of context, they do not validate the results.
1-The authors introduce a statistical process to justify, (validate) the results, however, the p-value could be equal 0.852 or 0.003 or p<0.001. What is value-added and how does it help the data validation? and it was not discussed.
P values are validated in method section and added below tables.
2-The presentation of the results in % must be preceded by the number, or vice-versa example in line 132
142 (n=28, 20%)
142 (20%, n=28)
Results section is revised according to suggestions.
Minor comments
1- Lines 132-136. these results expressed in table 1 must be indicated
Revised.
2-Line 137- "The majority of ColR-Kp belonged to ST101 (56%) and ST395(11%) clones" All 142 strains were ColR-Kp? so 20% were adult patients and 80% were children?
Yes, all 142 isolates were ColR-Kp, and the median age is 61 (0-91).
3-Lines 149-150 "significantly higher (95%, p=0.003)" and "very low (31%, p=0.002)" could you explain the meaning of p-values?
Revised and corrected as suggested.
4- Lines 150-151-Are there three ST395 strains with two carbapenemases?
Yes, revised and explained in the main text.
5- Line 191 "was magnified in images I-L" must be R-U
Corrected as suggested.
6- lines200-201 "One of the most promising strategies is inhibition of virulence factors of bacterium. one of the examples mentioned in major comments
Revised.
Lines 209-210 "Holden et al. demonstrated that during pneumonia siderophores stabilizes HIF-1 and increases bacterial dissemination to the spleen [14]. what is the meaning of HIF-1? and explain this sentence.
Revised and explained in detailed in the main text.
lines 211-212 "Lawlor et al. reported that the acquisition of yersiniabactin is an important step in the evolution of virulent K.pneumoniae [2,24]" and Bachman, M.A. [24]!!!!
The reference number 24 is removed from this part.
Lines 226-227 "ColR-Kp could be explained by their resistance inside neutrophils after being uptake". explain this sentence.
Revised in the main text.
Line240- "Branzk et al. reported that NETs are formed in response to large pathogens" is similar to title of article!!! explain this sentence.
Revised in the main text.
Reviewer 2 Report
In the paper “Virulence determinants of colistin-resistant K.pneumoniae High-Risk Clones”, by Dogan et al., the authors hypothesize that high-risk clones of colistin-resistant K.pneumoniae (ColR-Kp) show an higher number of virulence factors and enhanced survival capacity against neutrophil activity. To demonstrate this, they analyze in 142 patients with invasive ColR-Kp infections, either the virulence genes of isolated strains or neutrophil response. The main findings were that yersiniabactin biosynthesis gene and ferric uptake operon associated gene were higher in ST101 and ST395, as well as neutrophil uptake of kfu+-ybtS+ ColR-Kp was higher than kfu--ybtS- 25 ColR-Kp.
Data seem promising, and support the proposed hypothesis, though, as the authors state in the discussion, to confirm the correlation between iron uptake gene positivity and neutrophil response we needs further experimental studies.
Author Response
Thank you for your comments. We will conduct new studies on this topic
Round 2
Reviewer 1 Report
Only few questions related to Table 1 that has not been clarified:
-How is the average age of 61 (0-91) if the number of patients is 142 adults?
-Why in text is mentioned 58% (n = 82) of the patients were male and in table is female, must be consistent!
-Could the authors explain the values not statistically significant for bacteremia, VAP and Being in ICU?
Author Response
Only few questions related to Table 1 that has not been clarified:
-How is the average age of 61 (0-91) if the number of patients is 142 adults?
It is revised in the main text .
-Why in text is mentioned 58% (n = 82) of the patients were male and in table is female, must be consistent!
It is revised as suggested.
-Could the authors explain the values not statistically significant for bacteremia, VAP and Being in ICU?
These data could not be explained with our findings. Further studies must be implicated as we mentioned in the conclusion section.